# *Streptococcus gordonii* Supragingival Bacterium Oral Infection-Induced Periodontitis and Robust miRNA Expression Kinetics

**DOI:** 10.3390/ijms25116217

**Published:** 2024-06-05

**Authors:** Chairmandurai Aravindraja, Syam Jeepipalli, William D. Duncan, Krishna Mukesh Vekariya, Shaik O. Rahaman, Edward K. L. Chan, Lakshmyya Kesavalu

**Affiliations:** 1Department of Periodontology, College of Dentistry, University of Florida, Gainesville, FL 32610, USA; aravindrchairman@ufl.edu (C.A.); sjeepipalli@dental.ufl.edu (S.J.); kvekariya@ufl.edu (K.M.V.); 2Department of Community Dentistry and Behavioral Science, College of Dentistry, University of Florida, Gainesville, FL 32610, USA; duncanw@ufl.edu; 3Department of Nutrition and Food Science, University of Maryland, College Park, MD 20742, USA; srahaman@umd.edu; 4Department of Oral Biology, College of Dentistry, University of Florida, Gainesville, FL 32610, USA; echan@dental.ufl.edu

**Keywords:** periodontal disease, miRNAs, NanoString analysis, transient miRNA expression, machine learning, *S. gordonii*

## Abstract

*Streptococcus gordonii* (*S. gordonii*, Sg) is one of the early colonizing, supragingival commensal bacterium normally associated with oral health in human dental plaque. MicroRNAs (miRNAs) play an important role in the inflammation-mediated pathways and are involved in periodontal disease (PD) pathogenesis. PD is a polymicrobial dysbiotic immune-inflammatory disease initiated by microbes in the gingival sulcus/pockets. The objective of this study is to determine the global miRNA expression kinetics in *S. gordonii* DL1-infected C57BL/6J mice. All mice were randomly divided into four groups (*n* = 10 mice/group; 5 males and 5 females). Bacterial infection was performed in mice at 8 weeks and 16 weeks, mice were euthanized, and tissues harvested for analysis. We analyzed differentially expressed (DE) miRNAs in the mandibles of *S. gordonii*-infected mice. Gingival colonization/infection by *S. gordonii* and alveolar bone resorption (ABR) was confirmed. All the *S. gordonii*-infected mice at two specific time points showed bacterial colonization (100%) in the gingival surface, and a significant increase in mandible and maxilla ABR (*p* < 0.0001). miRNA profiling revealed 191 upregulated miRNAs (miR-375, miR-34b-5p) and 22 downregulated miRNAs (miR-133, miR-1224) in the mandibles of *S. gordonii*-infected mice at the 8-week mark. Conversely, at 16 weeks post-infection, 10 miRNAs (miR-1902, miR-203) were upregulated and 32 miRNAs (miR-1937c, miR-720) were downregulated. Two miRNAs, miR-210 and miR-423-5p, were commonly upregulated, and miR-2135 and miR-145 were commonly downregulated in both 8- and 16-week-infected mice mandibles. Furthermore, we employed five machine learning (ML) algorithms to assess how the number of miRNA copies correlates with *S. gordonii* infections in mice. In the ML analyses, miR-22 and miR-30c (8-week), miR-720 and miR-339-5p (16-week), and miR-720, miR-22, and miR-339-5p (combined 8- and 16-week) emerged as the most influential miRNAs.

## 1. Introduction

Multispecies microbial commensal communities, rather than single pathogens, in oral cavities are critical for the initiation and development of polymicrobial oral diseases (PD and caries). *S. gordonii*, a member of the oral mitis group of streptococci *S. gordonii*, is the most abundant first inhabitant of the human oral cavity and dental plaque. Acquired shortly after birth, it plays an important role in the assembly of the oral microbiota [1]. This commensal, non-pathogenic bacterium present in the skin, oral cavity, upper respiratory tract, and intestines has been strongly implicated in infectious complications such as apical PD, bacteremia, infective endocarditis, empyema, perihepatic abscesses, pyogenic spondylitis, or spondylodiscitis [2,3,4,5,6]. *S. gordonii* is an early initial bacterial colonizer, commensal supragingival Gram-positive bacterium that possesses cell surface proteins that can bind to the cell surface components of host mammalian cells, thereby producing hydrogen peroxide (H_2_O_2_) as a by-product of sugar metabolism, rapidly attaching to tooth surfaces and heart valves, and promotes the formation of plaque biofilms [7,8,9]. The cell wall components (lipoteichoic acids, teichoic acids, lipoproteins, serine-rich repeat adhesins, peptidoglycans, and cell wall proteins) are recognized by individual host receptors and are involved in virulence and immunoregulatory processes which cause host inflammatory responses [10]. This bacterium facilitates late bacterial colonizers such as *Porphyromonas gingivalis* and *Fusobacterium nucleatum* through a process of coaggregation [11,12]. Periodontitis is a polymicrobial dysbiotic chronic inflammatory disease caused by microbes, including bacteria, viruses, and fungi interacting in the host subgingival sulcus/pockets. A recent meta-analysis encompassing 50 studies comparing the prevalence of microorganisms other than the already-known periodontal pathogens in subgingival plaque and/or saliva samples between subjects with PD and periodontal health that showed 25 bacterial species, including *S. gordonii*, that were significantly associated with periodontitis [13].

MiRNAs are a class of short, small (21- to 25-nucleotide), non-coding RNAs important for post-transcriptional regulators to the gene expression in metazoans. More than 2000 human microRNAs from the miRBase showed links to predicted and validated targets [14]. These regulatory small miRNAs play critical roles in the regulation of gene expression and have potential as biomarkers [15,16,17,18]. Understanding their cellular functions is fundamentally significant for understanding disease pathogenesis.

Current knowledge is limited regarding how the oral commensal supragingival bacterium *S. gordonii* affects miRNA expression during PD induced by oral infection. MiRNAs are also strongly considered as diagnostic/genetic novel biomarkers for several diseases such as cardiovascular disease, HIV, diabetes, hypertension, several types of cancers, Alzheimer’s disease (AD), and autoimmune diseases. Moreover, the deregulation of miRNAs has been implicated in a multitude of diseases, and is known to regulate the immune response of hosts infected with bacteria (*Treponema pallidum*, *Helicobacter pylori*, *Mycobacterium avium*, *M. tuberculosis*, *Salmonella*, and *Listeria monocytogenes*). MiRNAs are resistant to degradation by ribonucleases and have been shown to be potential biomarkers for disease prediction and diagnosis. Accordingly, an improved understanding of the biogenesis pattern of miRNAs could potentially lead to the development of novel diagnostic biomarkers for polymicrobial infection-driven inflammatory diseases [19]. *S. gordonii* increased the expression of miR-4516 and miR-663a in the human oral epithelial cells, modulating validated targets of chemokine-associated pathways [20].

The inflammatory miRNAs miR-21 [21], miR-325-3p [22], miR-27a, miR-34a [23], miR-132 [24,25], miR-146a, miR-155 [26,27], miR-126-3p [28,29] miR-200, and mmu-let-7a [23] have been reported in the initiation and progression of PD. Recently, we characterized alterations in sex-specific microRNA after partial human mouth microbes (PAHMM) in an ecological time-sequential polybacterial periodontal infection (ETSPPI) mouse model and identified seven miRNAs (miR-9, miR-148a, miR-669a, miR-199a-3p, miR-1274a, miR-377, and miR-690) in both sexes that may be implicated in the pathogenesis of PD [19]. The five miRNAs (miR-195, 31, 125b-5p, 15a, and 423-5p) that were DE during the *P. gingivalis* monobacterial infection [30] and the seven miRs (miR-22, 486, 126-3p, 378, 151a-3p, 423-5p, and 221) during *T. denticola* monobacterial infection were also DE in chronic PD with diabetes [25,30]. Similarly, we recently identified *T. forsythia*-specific miRNAs (miR-let-7c, miR-210, miR-146a, miR-423-5p, miR-24, miR-218, miR-26b, and miR-23a-3p) and these miRNAs have also been reported in the gingival tissues and saliva of PD patients [23]. Further, several DE miRNAs that are significantly upregulated (e.g., miR-101b, miR-218, miR-127, and miR-24) are also associated with many systemic diseases such as atherosclerosis, AD, RA, osteoarthritis, diabetes, obesity, and several cancers [23]. These miRNAs are associated in induction of PD and their biogenesis pattern may be bacterial-specific and time dependent [23]. Furthermore, there is a lack of research on miRNA profiling throughout the progression of PD with *S. gordonii* linked with dental caries and PD. Therefore, to elucidate the changes in miRNA expression associated with *S. gordonii*-specific bacterial infection and subsequent PD, this study employed *S. gordonii* to induce experimental PD at two different time points.

A recent literature search revealed an absence of studies employing live *S. gordonii* to examine miRNA signature patterns under both in vitro and in vivo conditions. Thus, identifying miRNA signature patterns specific to *S. gordonii* infection is essential to understanding the complex inflammatory pathways initiated by *S. gordonii* during infection. Furthermore, we employed the ML algorithms XGB, RFC, LR, SVC, and MLP to the *S. gordonii* infection-induced miRNA data. In a recent study we published, two ML models, XGB and RFC, were employed to analyze periodontal miRNA in response to *T. forsythia* in the mouse model [23]. This approach provided deeper insights into the specific miRNAs associated with PD infection.

## 2. Results

### 2.1. Chronic Infection of S. gordonii Effectively Colonized in Mice Gingival Surface

We previously reported that the oral inoculation of Gram-negative subgingival periodontal bacteria (*P. gingivalis*, *T. denticola*, or *T. forsythia*) promotes bacterial colonization on the gingival surface and induces miRNA expression [19,23,25,30,31]. In the present study, new experiments were conducted designed to examine whether Gram-positive *S. gordonii*, inhabiting the supragingival region can colonize gingival epithelium and induce PD in mice. Gingival plaque samples of mice showed the presence of 40% *S. gordonii* DNA after the first infection cycle in both the 8-week and 16-week study groups. All 90% of the mice tested positive for the *S. gordonii*-specific 16S rRNA gene during the second infection cycle and 100% tested positive after the sixth cycle. This confirmed *S. gordonii* colonization in both 8 and 16 weeks (Table 1). We estimated *S. gordonii*-specific IgG antibody kinetics at each time point.

### 2.2. Higher Alveolar Bone Resorption (ABR) and Bacterial Dissemination to Distal Organs

Osteoclast activation by inflammatory cytokine sensitization and pathogenic factors of periodontal bacteria contributes to ABR in periodontal infection [32]. Microscopic images of mandibles from both sham-infected and *S. gordonii*-infected are shown in Figure 1B. Mice infected with *S. gordonii* at both 8 weeks and 16 weeks showed significantly higher ABR in the mandible (lingual) *p* < 0.05 (Adjusted *p*-value= 0.0004) for the 16-week group (Figure 1B,C). Similarly, a significantly higher (*p* < 0.0001) ABR was observed in maxilla palatal for 16-week infected mice. Aggressive gingival infections will permeate bacteria into the bloodstream and can settle on the heart lining and valves [33]. None of the aliquoted internal organs (heart, lungs, brain, liver, kidney, or spleen) tested positive for *S. gordonii* genomic DNA at both 8 and 16 weeks of infection (Appendix A).

### 2.3. NanoString Analysis of miRNAs in S. gordonii-Infected Mandibles

The NanoString platform is highly reliable for working with multiple sample types. It is an amplification-free technology that uses molecular barcodes to directly quantify RNA molecules without the need for reverse transcription. The RNA extracted from mandibles was analyzed for global miRNA profiling in mice infected with *S. gordonii* for 8 and 16 weeks (Table 2). nCounter miRNA expression profiling showed 191 upregulated miRNAs including miR-375, miR-34b-5p, miR-142-5p, miR-135a (Table 3 and Appendix A), and 22 downregulated miRNAs including miR-133a, miR-1224, miR-2135, miR-499 (Appendix A) in 8 weeks of *S. gordonii*-infected mandibles compared to sham-infected mandibles. During 16 weeks of infection, 10 miRNAs (miR-1902, miR-203, miR-98, miR-210) were found to be upregulated (Table 4), and 32 miRNAs (miR-720, miR-1937c, miR-2135, miR-326) were downregulated (Appendix A). A *p*-value of <0.05 and a fold change of 1.1 and above was considered for analysis and significance (Table 3). Both upregulated and downregulated DE miRNAs between 8 and 16 weeks are shown in Appendix A.

### 2.4. Identification of Differentially Expressed (DE) miRNAs

A total of 40 samples from the four infection and sham infection groups were analyzed using high-throughput nCounter^®^ miRNA Expression Panels. To determine the DE miRNAs with statistical significance, we performed a volcano plot analysis, where we plotted log2 fold change on the *x*-axis and the negative log of *p*-value on the *y*-axis. We identified 19 downregulated (red) miRNAs and 19 upregulated (green) miRNAs that displayed a fold difference of +1.1 with a *p*-value of <0.05 in 8-week *S. gordonii*-infected mice, compared to the 16-week infected group (Figure 2A). The black dots represent miRNAs that do not pass the filter parameters. A total of 191 miRNAs showed higher expression during 8 weeks of infection (e.g., miR-375, miR-34b-5p, miR-142-5p, and miR-135a) and 10 miRNAs (e.g., miR-1902, miR-203, miR-210, miR-876-3p) were showed higher expression during 16 weeks of *S. gordonii*-infected mandibles (Figure 2B, Table 3 and Table 4). The miRNAs upregulated for 8 weeks are associated with the differentiation of human periodontal ligament stem cells (miR-375), acute inflammation, injury in the lung (miR-34b-5p), inflammation in the brain (miR-142-5p), lowered (miR-135a) in osteoblastic differentiation and higher (miR-323-3p) in obese PD patients. During a 16-week infection of *S. gordonii*, certain miRNAs were found to increase in response to lipoteichoic acid stimulation (miR-1902), in periodontal diabetic patients with gingivitis (miR-203), in obese PD patients (miR-210), and in PD patients in general (miR-423-5p). The expression of miR-323-3p is consistent across the *S. gordonii* DE, XGB, and RFA analyses along with the expression of DE-miRNAs miR-142-5p and miR-1968, as identified by XGB and RFA, respectively.

**Table 3 ijms-25-06217-t003:** *S. gordonii* infection-induced upregulated miRNAs, reported functions, and target genes.

Upregulated miRNAs in 8-Week *S. gordonii* Infection				
miRNA	Fold Change	*p*-Value	Reported Function	Number of Target Genes
miR-375	2.68	0.0079	Facilitated the proliferation and osteogenic differentiation capacity in human periodontal ligament stem cells [33]. Potential biomarker in esophageal cancer [15]. Upregulated in the saliva of migraine without aura patients [34].	27 (e.g., *Akap2, Atrn, Lpin1, Rpsa, Pacsin1*)
miR-34b-5p	1.76	0.0292	Upregulated in acute lung injury [35]. Regulates the lung inflammation, and apoptosis in LPS-induced ALI mouse model [36]. Upregulated in the saliva of migraine without aura patients [34].	354 (e.g., *2310010M20Rik, Rg9mtd2, Dnase2a, Epb4.2, Fyco1*)
miR-142-5p	1.58	0.0002	High levels were reported in the condition of experimental autoimmune encephalomyelitis mice [37]. Promoting brain inflammation through astrocyte activation in traumatic brain injury [38].	24 (e.g., *Abcg1, Cd28, Sort1, Slc35a1, Cwc25*)
miR-135a	1.56	0.0056	Inhibiting the differentiation of osteoblasts [39], and regulating osteoblastic activity [40]. Serving the role in osteoporosis progression by regulating osteogenic differentiation via RUNX2 [41].	19 (e.g., *Nup133, Adamts14, Cep170, App, Smad5*)
** miR-323-3p **	1.55	0.0011	Upregulated in obese individuals with periodontitis [42].	25 (e.g., *Fam190a, Eed, Fzd3, Pira1, Tyw1, Ccser1*)
miR-485	1.45	0.0015	up-regulated in osteoporosis patients [43]. Dysregulated miR-485 is associated with AD and Parkinson’s disease [44].	14 (*Cd93, Pigm, Fasl, Gns*)
miR-881	1.44	0.0007	Upregulated in dental pulp cells induced by 5-AZA-CdR [45]. Upregulated in the NAFLD liver tissue [46]. Downregulated in the chronic alcoholic rats [47]. Upregulated in the acute lung injury mice [48].	50 (e.g., *Xiap, Stxbp5, Slc31a2, Wbp1l*)
miR-669e	1.43	0.0004	Downregulated in the flavonoid-treated mice [49].	108 (e.g., *Wdr59, Mmp25, Ubac1, Mzt1, Fzd2*)
miR-187	1.43	0.0004	Upregulated in the sepsis-induced myocardial dysfunction- mice supplemented with mesenchymal stem cells [50]. Reported as a diagnostic and predictive factor for oral squamous cell carcinoma patients [51]. Restoring the osteogenic differentiation capacity in bone marrow-derived mesenchymal stem cells [52]. The most abundant miR in the serum of women in different stages of pregnancy or patients with different urothelial cancers [53].	14 (e.g., *Cd93, Dmpk, Rgs4, Rnaseh2a, Adcy1*).
miR-1306	1.43	0.0027	Mediating the feedback regulation of the TGF-β signaling pathway in granulosa cells [54]. Regulating the human ameloblastoma AM-1 cells differentiation [55].	22 (e.g., *Rinl, Frmd3, Tpmt, Ptprb, Map3k7*).
** miR-200b **	1.43	0.0253	Variations in the levels of miR-200b were observed in gingival tissue of obese periodontitis subjects [42].	55 (e.g., *Zeb2, Suz12, Bmi1, Flt1, Mapk14*)
miR-767	1.42	0.0076	Upregulated in hepatocellular carcinoma [56]. Forced upregulation of miR-767-5p may represent a novel therapeutic strategy for glioma patients by targeting SUZ12 [57]. Highly expressed in senescent vascular endothelial cells [58].	13 (e.g., *Mcl1, Gtpbp1, Klf6, Pigs, Tmem178b*)
miR-455	1.41	0.0006	Associated with cartilage development, adipogenesis, preeclampsia, and cancers in humans [59].	31 (e.g., *Acvr2b, Smad2, Chrdl1, Fut1, Cd36*)
miR-2140	1.4	0.0007	Predictive markers for the onset of cognitive impairment [60]. Upregulated in the differentiated P19 embryonic carcinoma cells [61].	------------
miR-1187	1.4	0.0027	Upregulated in hypoxia-induced apoptotic HL-1 cardiomyocytes [62]. Functions as a negative regulator of osteogenesis [63]. Upregulated in high glucose-treated kidney podocytes, and kidney tissue in db/db mice [64]. Regulates hepatocyte apoptosis by targeting caspase-8 [65].	148 (e.g., *Ado, Lifr, Igf2, Zfp157, Slc39a14*)
miR-291a-3p	1.4	0.0039	Improve cell viability and osteogenic differentiation of BMSCs [66]. Decreased expression was reported in Cigarette smoke components exposed rat models [67] and in the lungs of LPS-induced ALI mouse models [68] and in spinal cord ischemia-reperfusion injury [69].	73 (e.g., *Cdkn1a, Dkk1, Pax6, Myl6b, Zfp568*)
miR-1968	1.4	0.0071	Downregulated in lupus nephritis mice [70]. Associated with hepatic energy metabolism [71].	19 (e.g., *Snx27, Creb5, Ptpre, Itgav, Cdc42bpa*)
miR-1957	1.39	0.0018	Expressed in the lophotrochozoan taxon [72]	12 (*Sorcs1, Il18r1, Tbc1d24, Pla2r1, Pin1*)
miR-129-3p	1.39	0.0025	Downregulated in periodontitis patients [73]. Reported as protective miR from cardiomyocyte hypertrophy [74]. Tumor suppressor miR in the ovarian cancer cells [75].	0
miR-1942	1.39	0.0029	Upregulated in tumors that evaded the prevention of immunotherapy [76].	14 (e.g., *Zfp760, Rapgef3, Il17ra, Alkbh1, Slc8a1*)
miR-2861	1.38	0.0006	Promoting osteoblast differentiation [77]. Downregulated in HPV-E6 overexpressed HEK293T cells and HaCaT cells [78]. Upregulated in papillary thyroid carcinoma with lymph node metastasis [79].	11 (e.g., *Hdac5, Runx2, Hlcs, Plxnc1, Cyb5r1*)
miR-671-3p	1.38	0.0006	Involved in the pathogenesis of atherosclerosis, rheumatoid arthritis, and acute myocardial infarction [80]. Reported as a positive regulator of glioma progression and development [81].	1 (*Pbx1*)
miR-290-5p	1.38	0.0078	Reported to suppress the breast cancer progression [82].	160 (e.g., *Adra2b, Birc5, Rplp0, Bach2, Chic1*)
miR-509-5p	1.38	0.017	Has a role in alleviating myocardial infarction damage [83]. Serving as a potential biomarker for evaluating osteosarcoma prognosis [84].	43 (e.g., *BC016495, Abcb7, Slc7a2, Cflar, Cxcr2*)
miR-205	1.38	0.0179	Downregulated in chronic periodontitis patients [85]. Downregulated in LPS-induced periodontal ligament stem cells [86]. Plays an important role in the physiology of epithelia via regulation of pathways governing differentiation and morphogenesis [87]. Reduces actomyosin contractility and activates hair regeneration in young and old mice [88].	44 (e.g., *Pten, Lrrk2, Dok4, Atp1a1, Gsk3b*)
miR-466d-3p	1.37	0.0022	Downregulated in infarction-exposed fetal and adolescent sheep heart tissue [89].	202 (e.g., *Rad51l3, Gcfc1, Fam54a, Tmem20, Chic1*)
miR-34a	1.37	0.0032	Potential therapeutic anticancer miRNA [90]. Downregulated in adolescent sheep heart infarct samples [89].	40 (e.g., *Dnase2a, Mapre1, Cd93, Mbd4, Fyco1*)
miR-2139	1.37	0.0039	Upregulated in UV-B-induced tumors [91].	1 (*Cd4*)
miR-34b-3p	1.37	0.0075	Significantly expressed in the mice’s spinal cord development [90].	11(e.g., *Raph1, Pbx1, Cops2, Sec61b, Ttc19*)

Details of the biological function and target genes were given for the 29 significantly DE miRNAs in 8-week infected mice mandibles out of 191 upregulated miRNAs, and other miRNAs shown in the Appendix A. miRNAs associated with periodontal disease are marked in red.

**Table 4 ijms-25-06217-t004:** *S. gordonii* infection-induced upregulated miRNAs, reported functions, and target genes.

Upregulated miRNAs in 16-Week *S. gordonii* Infection				
miRs	Fold Change	*p*-Value	Reported Function	Number of Target Genes
miR-1902	1.88	0.0005	Overexpressed in mice serum and whole blood after intraperitoneal injection of lipoteichoic acid (LTA) [92]	5 (e.g., *Evi2b, Arhgef15, Itga11, Myo9a, Tsn*)
miR-203	1.47	0.0012	Upregulated in the human periodontal disease [27]. Elevated in diabetic periodontal patients [93]. Elevated in primary gingival epithelial cell infection with *P. gingivalis* 33277 [94]. Regulating the periodontal ligament cells—stimulated by LPS of *P. gingivalis* bacteria [95]. Master modulator and fine-tuning neuro-inflammation [96]. Decreased expression is reported in periodontitis and is consistent with increased angiogenesis in periodontitis [97].	70 (e.g., *Stxbp4, Trp63, Zfp281, Snora62, Cav1, Vcan*)
miR-98	1.42	0.0013	Overexpression protects the cardiomyocytes against apoptosis [98]. Critical miRNAs are regarded as a potential biomarker candidate for Crohn’s disease [16]	39 (e.g., *Acvr1b, Mmp11, Il6, Phka1, Poteg*)
miR-210	1.39	0.0105	Elevated in obese periodontitis patients [99]. Human periodontal ligament stem cells had elevated miR-210 in the presence of Endobon-xenograft granules [100].	47 (e.g., *Tcf7l2, Acvr1b, Ucp2, NFKB1, Bcl2*)
miR-876-3p	1.29	0.01	miR-876-3p suppresses the progression of colon cancer and correlates with the prognosis of patients [101].	19 (e.g., *Sorcs1, Ccsap, Plekha2, Il18r1, Fam161b*)
mmu-let-7c	1.29	0.0305	Interfere with critical inflammatory cytokine production viz., IL-1β, IL-6, and TNF-α in human osteoarthritis (OA) and RA [102]. Playing a role in cardiomyogenesis promotion activity [103].	37 (e.g., *Sall4, Myc, Lin28a, EZH2, Gnl3l*)
miR-423-5p	1.25	0.0165	Upregulated expression in severe periodontal disease [104]. Higher expression in obese periodontitis subjects [105]. Identified as a new candidate biomarker in the cross-talk between diabetes mellitus and AD [106].	2 (e.g., *Map1b, Zmat3*)
miR-361	1.21	0.0257	miR-361-3p/Nfat5 signaling axis controls cementoblast differentiation [107].	31 (e.g., *Eea1, Dnmt3a, Ctbp2, Vps26a, Zfp120*)
mmu-let-7a	1.19	0.0343	A proinflammatory role for let-7 miRNAs in experimental asthma [108].	
miR-101b	1.16	0.021	Increased expression promotes apoptosis of endothelial cells in acute coronary syndrome [109]. Major mediator of tauopathy and dendritic abnormalities in AD progression [110].	44 (e.g., *Stc1, Atxn1, Map7d1, Rcor3, Msi2*)

Details of the biological function and target genes were given for the top ten significantly expressed miRNAs in 16-week infected mice mandibles. miRNAs associated with periodontal disease are marked in red.

### 2.5. DE miRNAs and Functional Pathway Analysis

We conducted functional enrichment analysis to predict the biological function of DE, both in the up-regulated and down-regulated miRNAs. The functional pathways associated with the altered upregulated miRNAs at 8 and 16 weeks of infection were assessed using DIANA-miRPath software (https://diana.e-ce.uth.gr/home). The KEGG pathway analysis revealed that the upregulated genes were significantly enriched in mitogen-activated protein kinase (MAPK) signaling, pathways in cancer, WNT (Wingless and Int-1) signaling pathway, Focal adhesion, TGF-beta signaling pathway, and Ubiquitin mediated proteolysis are commonly identified in DE miRNAs of 8 and 16 weeks (Figure 2C). The other important pathways linked with 8 weeks of infection were bacterial invasion of epithelial cells, B-cell receptor signaling pathway, and T-cell receptor signaling pathways. According to KEGG pathway enrichment analysis, the DE miRNA-target genes were significantly enriched in pathways of bacterial invasion of epithelial cells (BIEC) pathway (8 weeks) and MAPK signaling pathway (16 weeks) *S. gordonii*-infected mandibles compared with sham-infected controls at *p* < 0.05. *S. gordonii* significantly altered 24 gene expressions based on an upregulated miRNA profile in the 8-week infection period (Figure 3). The binding ability of miRNA and its target genes were analyzed using MiRTarBase for both 8- and 16-week infection. It has accumulated more than 300 and 60,000 miRNA-target interactions (MTIs) that were validated by the reported assay, Western blot, microarray, and next-generation sequencing experiments. Each miRNA has a different target gene with a specified MiRTarBase ID (Appendix A).

### 2.6. Machine Learning Analysis of miRNA Copies

Our ML analysis results are divided into two groups: tree-based methods consisting of XGB and RFC, and non-tree-based methods consisting of LR, SVC, and MLP. With the 8-week dataset, mmu-miR-30c had the highest impact on the tree-based models (Figure 4A, XGB; Figure 5A, RFC), and mmu-miR-22 had the highest impact on the non-tree-based models (Figure 6A, LR; Figure 7A, SVC; Figure 8A, MLP). For the 16-week dataset, mmu-miR-339-5p had the highest impact on the tree-based models (XGB, RFC; Figure 4B and Figure 5B), while mmu-miR-720 had the most impact on the non-tree-based models (LR, SVC, MLP; Figure 6B, Figure 7B and Figure 8B). Finally, for the combined 8- and 16-week dataset, mmu-miR-339-5p was again the most impactful for the tree-based models (XGB, RFC; Figure 4C and Figure 5C), and mmu-miR-22 and mmu-miR-720 had the highest impacts for the non-tree-based models (LR, SVC, MLP; Figure 6C, Figure 7C and Figure 8C). The results of the SHAP value analysis are summarized in Figure 4, Figure 5, Figure 6, Figure 7 and Figure 8, and a list containing the importance of each miRNA as predicted by the RFC and XGB model is summarized in Table 5. Descriptions of the miRNA functions for the most important agreed-upon features are provided in Table 5. Table 5 provides descriptions for the tree-based models, and Appendix A provides descriptions for the non-tree-based models (i.e., LR, SVC, and MLP). The List of miRNAs unique in expression among the DE and 5 ML models depicted in Appendix A.

In addition to determining which miRNAs were identified by each model, we analyzed which miRNAs had two or more models that agreed for both the miRNA and level of importance (Table 6). The highest agreed-upon miRNAs are consistent with the results mentioned earlier (i.e., in the previous paragraph), but we also found high levels of agreement in the miRNAs ranked 2–5. For the tree-based models, only miR-142-5p was determined as the fifth most important feature in the combined 8- and 16-week datasets (Table 5). For the non-tree-based models, we found more agreement. For the second-ranked feature, LR, SVC, and MLP agreed on miR-1 for the 8-week dataset; SVC and MLP agreed on miR-1 for the 16-week dataset; and LR, SVC and MLP agreed on miR-22 for 8- and 16-week dataset (Table 6). For the third-ranked feature, LR and MLP agreed on miR-720 for the 8-week dataset; LR and SVC agreed on miR-205 for the 16-week dataset; and LR, SVC, and MLP agreed on let-7a for the 8- and 16-week datasets. For the fourth-ranked feature, the only agreement was let-7c in the 8- and 16-week dataset for LR, SVC, and MLP. For the fifth-ranked feature, SVC and MLP agreed on let-7c for the 16-week dataset, and LR and SVC agreed on miR-339-5p for the 8- and 16-week datasets (Table 6). These results are summarized by the cells with the gray background in Table 7.

Lastly, we analyzed which miRNAs we agreed upon by at least two models regardless of rank. These results largely overlap with analysis agreement for both the miRNA and level of agreement, but also include the following miRNAs. For the 8-week dataset, LR (rank 4) and SVC (rank 5) agreed upon let-7a, and RFC (rank 2) and XGB (rank 3) agreed on miR-323-3p (Table 7). For the 8- and 16-week datasets, RFC (rank 3) and XGB (rank 4) agreed on miR-323-3p. We also found agreements of a single model that disagreed with the others for the same miRNA. For the 16-week dataset, LR (rank 4) also agreed with SVC and MLP for miR-1, and MLP (rank 4) also agreed with LR and SVC for miR-205. For the 8- and 16-week dataset, LR (rank 1) agreed with SVC and MLP for miR-22, and LR (rank 2) agreed with SVC and MLP for miR-720. These results are summarized by the cells without a gray background in Table 7.

**Table 5 ijms-25-06217-t005:** Random Forest classifier and XGBoost most importantly agreed upon miRNA features and reported miRNA functions.

miRNA	MIMAT #	Target Functions
**8-Week Analysis**		
miR-30c	MIMAT0000514	Suppressing the osteogenic differentiation in the human periodontal ligamental stem cells [111]. miR-30 family had a role in the occurrence and development of bone and joint disease [112]. Serving as a non-invasive biomarker for early oral squamous cell carcinoma [17]. Upregulated in multiple system atrophy compared with Parkinson’s disease and healthy subjects [18]. Decreased expression was observed in fasting and post-prandial period of high post-prandial response [113].
miR-323-3p	MIMAT0000551	Highly expressed in blood samples from patients with coronary heart diseases and rat models of CHD [114].
**16-Week Analysis**		
**miRNA**	**MIMAT #**	**Target Functions**
miR-339-5p	MIMAT0000584	Most predictive periodontal miRNA in the *T. forsythia*-induced mice periodontitis [23]. Downregulated in human atherosclerotic plaques and ox-LDL-induced cells [115]. Downregulated in cardiac tissue of 7-day-old mice [116].
**Combined 8 and 16 Weeks**		
**miRNA**	**MIMAT #**	**Target Functions**
miR-339-5p	MIMAT0000584	Shown in the 16-week analysis
miR-323-3p	MIMAT0000551	Shown in the 8-week analysis.
miR-142-5p	MIMAT0000154	Upregulated in human gingival crevicular fluids [117]. Regulates inflammation in multiple sclerosis mice models [37]. Decreased expression was reported in breast cancer [118]. Increased expression was reported in macrophages from the tissue samples of patients with liver cirrhosis and idiopathic pulmonary fibrosis [119].

miRNAs associated with periodontal disease are marked in red.

**Table 6 ijms-25-06217-t006:** Summary of the agreed-upon importance (i.e., rank) and miRNA for each machine learning model.

		miRNA Feature Rank				
Dataset	miRNA	1	2	3	4	5
8 weeks	miR-22	LR, SVC, MLP				
	miR-1		LR, SVC, MLP			
	miR-720			LR, MLP		
	let-7a				LR	SVC
	miR-23a					LR
	miR148a				SVC	
	let-7c			SVC		
	miR-125b-5p				MLP	
	miR-199a-3p					MLP
	mR-30c	RFC, XGB				
	miR-340-3p		XGB			
	miR-323-3p		RFC	XGB		
	miR-499				XGB	
	miR-449b			RFC		XGB
	miR-29a				RFC	
	miR-291b-5p					RFC
16 weeks	miR-22		LR	MLP		
	miR-720	LR, SVC, MLP				
	let-7c					SVC, MLP
	miR-1		SVC, MLP		LR	
	miR-205			LR, SVC	MLP	
	miR-126-3p					LR
	miR-133a				SVC	
	miR-339-5p	RFC, XGB				
	miR-325		XGB			
	miR-503			XGB		
	miR-711				XGB	
	miR-343					XGB
8 and 16 weeks	miR-22	LR	SVC, MLP			
	miR-720	SVC, MLP	LR			
	let-7a			LR, SVC, MLP		
	let-7c				LR, SVC, MLP	
	miR-1					LR, SVC
	let-7g					MLP
	miR-339-5p	RFC, XGB				
	miR-345-3p		RFC			
	miR-323-3p			RFC	XGB	
	miR-m59-2				RFC	
	miR-142-5p					RFC, XGB
	miR-342-5p		XGB			
	miR-450a-5p			XGB		

LR—Logistic regression, SVC—C-support vector classifier, MLP—multilayer perceptron, RFC—random forest classifier, and XGB—extreme gradient boosting.

**Table 7 ijms-25-06217-t007:** Summary of the agreements for the miRNA and importance (i.e., rank) between each machine learning model.

		miRNA Feature Rank				
Cohort	miRNA	1	2	3	4	5
8 weeks	miR-22	LR, SVC, MLP				
	miR-1		LR, SVC, MLP			
	miR-720			LR, MLP		
	let-7a				LR	SVC
	mR-30c	RFC, XGB				
	miR-323-3p		RFC	XGB		
16 weeks	miR-720	LR, SVC, MLP				
	let-7c					SVC, MLP
	miR-1		SVC, MLP		LR	
	miR-205			LR, SVC	MLP	
	miR-339-5p	RFC, XGB				
8 and 16 weeks	miR-22	LR	SVC, MLP			
	miR-720	SVC, MLP	LR			
	let-7a			LR, SVC, MLP		
	let-7c				LR, SVC, MLP	
	miR-1					LR, SVC
	miR-339-5p	RFC, XGB				
	miR-323-3p			RFC	XGB	
	miR-142-5p					RFC, XGB

Cells with a gray background identify cases in which the two more models agreed on both the importance and miRNA.

## 3. Discussion

The sex-specific DE miRNA reported in the partial human mouth microbes (PAMHH)- ETSPPI model that utilized five different bacteria including *S. gordonii* (early bacterial colonizer), *F. nucleatum* (intermediate bacterial colonizer), *P. gingivalis, T. denticola* and *T. forsythia* (late bacterial colonizers) using a time sequential infection [19]. Recently, we reported a polybacterial PD mouse model that induced miRNA expression, but the novelty of studying specific bacterium *S. gordonii* alone in inducing PD and specific miRNAs DE was not examined. Specific signature DE miRNAs in *P. gingivalis* [30], *T. denticola* [25] and *T. forsythia* [23] monoinfections were reported. We examined *S. gordonii* bacterium colonization on the mouse gingival surface, horizontal ABR measurements, intravascular dissemination of *S. gordonii* to distal organs, and global miRNA profiling in *S. gordonii* mice infected intraorally at 8 weeks and 16 weeks. Mice infected with *S. gordonii* at both time points indicated 100% bacterial colonization on the gingival surface, which was documented with 16S rRNA gene amplification. Similarly, *S. gordonii* significantly induced higher ABR in mice at both 8- and 16-week infection time-points. To the best of our knowledge, this is the first report that live *S. gordonii* a supragingival Gram-positive bacterium, significantly colonized the gingival epithelium, and induced higher ABR during intraoral infection. These results are in accordance with the report that *S. gordonii* in the periapical lesions of patients with apical periodontitis may contribute to the induction of ABR [3]. Among the 201 DE upregulated miRNAs in 8-week and 16-week *S. gordonii*-infected mice, eight of the miRNAs were shown to be observed in human periodontitis patients indicating its robust miRNA genesis in induction of PD (miR-323-3p, miR-200b, miR-129-3p, miR-205, miR-203, miR-210, miR-423-5p, and miR-146b). Four of the DE miRNAs, miR-187, miR-671-3p, miR-509-5p, and miR-767 were associated with cardiovascular diseases. Five DE miRNAs, miR-135a, miR-187, miR-1187, miR-291a-3p, and miR-2861 were also reported in the osteocyte regulation. Two of the miRNAs, miR-485 and miR-101b, are associated with Alzheimer’s disease. All of the 39 DE miRNAs were reported to be observed in *S. gordonii*-induced periodontitis. The reviewed observations of these miRNAs in the human periodontal gingival tissues, *S. gordonii*-induced PD in mice models highlight the importance of oral bacterial species normally associated with oral health to understand the present reported miRNAs as potential biomarkers for *S. gordonii*-induced periodontitis.

The inflammatory miRNA (miR-146b) reported in the PD subjects is also upregulated in the 8- and 16-week *S. gordonii*-infected mice [120]. Thirty-three miRNAs (e.g., miR-375, 148a, 574-3p, 382, and 181c) DE in the polymicrobial infection [19] are also expressed in the present *S. gordonii* monobacterial infection (Female comparison). Twenty-three miRNAs (e.g., miR-200b, 2141, 202-5p, and 1902) DE in the polymicrobial infection [19] are also expressed in the present *S. gordonii* monobacterial infection (Male comparison). The seven miRNAs (miR-31, 152, 423-5p, 191, 103, 30d, and 30c) that were DE during the *P. gingivalis* monobacterial infection [30] are observed in the present *S. gordonii* monoinfection. The five miRNAs (miR-2135, 30c, 126-5p, 423-5p, and 101b) reported in the *T. denticola* monobacterial infection [25] are also observed in the present *S. gordonii* monoinfection. The eight miRNAs (miR-1902, let-7c, 423-5p, 210, let-7a, 98, 876-3p, and 101b) reported in the *T. forsythia* monoinfection [23] are also induced in the present *S. gordonii* monoinfection. These data indicate that most of the periodontal-causing bacteria rely on the common miRNA activation pathways and initiate periodontitis.

The DE upregulated miRNAs (out of 191, only 20 miRNAs used in pathway analysis) in the 8-week *S. gordonii* infection involved in the pluripotency of stem cells pathways, MAPK-signaling pathway, Axon guidance, Wnt signaling pathway, Arrhythmogenic right ventricular cardiomyopathy, mTOR signaling pathway, and thyroid hormone signaling pathway. Among these pathways, bacterial invasion of epithelial cells was highlighted showing the number of genes altered by the associated 11 miRNAs. The DE upregulated miRNAs in the 16-week *S. gordonii* infection involved in axon guidance pathways, MAPK signaling pathway, Wnt signaling pathway (bone formation), protein digestion and absorption pathways, and focal adhesion pathways.

Interestingly, miRNA-375 is a novel multifunctional regulator miRNA that interacts with a large number of 27 target genes and is involved in the regulation of the differentiation and functioning of cells of the nervous and immune systems, bone and adipose tissue, and development and virus (HPV, HBV, and HIV) replication [121]. Anomalous expressions of miR-375 are found in multitudinous diseases, including PD during polymicrobial infections, *T. denticola, T. forsythia*, and *S. gordonii* (8 and 16 weeks) monoinfections, carcinogenesis, inflammation, and autoimmune and cardiovascular diseases. miR-375 is one of the most frequently down-regulated miRNAs in esophageal cancer [122], H. pylori-induced gastric carcinogenesis, and AIDS-KS patients after cART [123]. Gut epithelium-expressed miR-375 has been shown as a key regulator of epithelial properties (gut mucosal immunity) that are necessary for securing epithelium-immune system crosstalk in miR-375-deficient mice [124]. A recent review stated that miR-375 attenuates PD-1/PD-L1 signaling [125], a significant immune checkpoint mentioned in the 2018 Nobel Prize in Physiology or Medicine, indicating that miR-375 can be studied as a latent immune checkpoint modulator in immune investigations [121]. The major limitation of the current study is the inability to use blood and salivary secretions in this microRNA analysis.

For the machine learning analysis, the differences between the tree-based models (RFC, XGB) and non-tree-based models (LR, SVC, MLP) are quite interesting. First, we did not find any overlap between the miRNAs determined to be important for each of these groups of models. Second, both the tree-based and non-tree-based models agreed on their most important miRNAs for the 8-week and 16-week datasets. However, for the 8- and 16-week datasets, we found that LR differed from SVC and MLP. The reasons for these differences need to be investigated.

It is also important to note that we did not include an in-depth discussion of the ML metrics for each model. Each model scored between 99–100% for accuracy, precision, recall, and F1 score. This almost certainly indicates that the models have overfit the data. The reason for this is that due to the small number of mice used in the study, it is very difficult (most likely impossible) to divide the study groups into meaningful subgroups that can be used for training and testing. Instead, we have focused on using ML to gain insights into which miRNAs may be most relevant in identifying infected mice. To meaningfully score our models, we would need to train and test our models on large cohorts of mice.

In a meta-analysis study, Li et al. analyzed and predicted miR-152-3p and miR-34a-5p neurological outcomes in patients with subarachnoid hemorrhage [126]. The studies of Chi et al. predicted 30 miRNA features in pancreatic cancer and Li et al. used RFA in predicting potential miRNAs in diabetes [127,128]. Multilayer perceptron was used to improve the prediction performance in the miRNA and disease associations in lymphoma (e.g., hsa-let-7a), leukemia (e.g., hsa-miR-218), kidney neoplasms (e.g., has-miR-494), colon neoplasms (e.g., has-miR-30e), and breast neoplasms (e.g., has-miR-106a) [129].

## 4. Materials and Methods

### 4.1. Animal Models and Ethics Statement

All the animal procedures were performed according to the guidelines (University of Florida Institutional Animal Care and Use Committee protocol number 202200000223). This preclinical mouse study complied with the ARRIVE guidelines (Animal Research: Reporting In Vivo Experiments). Forty 8-week-old wild-type C57BL/6 mice, containing 20 males and 20 females used. The animals housed in a controlled temperature environment with 12 h dark/light cycles and access to mice chow and water ad libitum. The sample size was determined based on our previous studies [19,23,25,30].

### 4.2. Bacterial Culture and Mice Oral Administration

*S. gordonii* (Sg) DL1 was grown in Brucella blood agar plates, supplemented with hemin and vitamin K (Hardy Diagnostics, Santa Maria, CA, USA) for three days in Coy anaerobic chamber [19,23,25,30]. A bacterial cell suspension was prepared by combining an equal volume of reduced transport fluid (RTF) and 6% carboxymethyl cellulose (CMC) for intraoral infection in mice [19]. Both genders of mice were divided randomly into four groups (*n* =10) (Group-I: Sg-infected for 8 weeks; Group II: Sg-infected for 16 weeks; Group-III: sham-infected for 8 weeks; Group-IV: sham-infected for 16 weeks) (Table 1). Before the bacterial infection, mouse oral microflora was suppressed with Kanamycin (500 mg) dissolved in two liters of sterile water for 3 days followed by rinsing with 0.12% of chlorhexidine gluconate ((Peridex: 3M ESPE Dental Products, St. Paul, MN, USA) [19,25,30]. After the kanamycin antibiotic washout period, mice in Groups I and II underwent infection with *S. gordonii* (10^8^ cells) as described previously [19,23,25,30]. Four bacterial oral infection cycles, as described, were performed for Group I (8 weeks) and eight infection cycles, as described [30], were performed for Group II mice (16 weeks). One infection cycle consists of four days per week of *S. gordonii* intraoral infection/vehicle control suspension for every alternative week (Table 1 and Figure 1A). An equal volume of RTF and CMC was used as a vehicle control for sham-infected mice (Groups III and IV). These sham-infected mice groups served as common sham-infection controls for 5 different monoinfection mice studies performed simultaneously [23,25,30]. At the end of 8 and 16 weeks, the mice were euthanized by the carbon dioxide (CO_2_) inhalation method; blood was collected by cardiac puncture and serum was separated. Additionally, distal organs such the brain, heart, liver, lungs, spleen, and kidneys were collected. The left maxilla and mandibles were also collected in RNAlater for miRNA analysis, whereas the right maxilla and mandibles were designated for morphometry analysis of horizontal alveolar bone levels [19,23,25,30].

### 4.3. Molecular Detection of Bacteria

Two days after each Sg-infection cycle, gingival plaque samples were collected from gingival surfaces of the mice using sterile cotton swabs and the swabs were submerged in Tris EDTA (TE) buffer [19,23,25,30]. Colony PCR was used to investigate the Sg bacteria in gingival plaque samples. Genomic DNA from the heart, lungs, liver, spleen, brain, and kidney were extracted following the standard procedure described in Qiagen DNeasy Blood and Tissue Kit(Qiagen, Germantown, MD, USA)and stored at −20 °C [19,23,25,30]. The presence of *S. gordonii* genomic DNA was confirmed using PCR with primers specific to the 16S rRNA gene as described previously [19,130]. The forward primer 5′-GTAGCTTGCTACACCATAGA-3′ and reverse primer 5′-CTCACACCCGTTCTTCTCTT-3′ used to amplify the *S. gordonii* DL1-16S rRNA gene [19]. DNA extracted from *S. gordonii* DL1 bacterial culture was used as positive control and sterile Milli-Q water was considered negative control in the PCR reaction. The PCR amplified products were run on 1% agarose gel electrophoresis and visualized under UVP GelStudio touch Imaging System (Analytik Jena US LLC, CA, USA).

### 4.4. Measurement of Alveolar Bone Resorption (ABR)

The impact of *S. gordonii* infection and sham infection on ABR was quantitatively assessed using histomorphometry, as described previously [19,23,25,30]. Briefly, the jaw specimens (mandibles and maxilla) were autoclaved and defleshed. H_2_O_2_ (3%) solution was used to clean the jaw bone and air dried [19]. Two-dimensional alveolar bone imaging was performed using a stereo dissecting microscope (Stereo Discovery V8, Carl Zeiss Microimaging, Inc., Thornwood, NY, USA). A line tool was used to measure the horizontal ABR between cementoenamel junction and alveolar bone crest (AxioVision LE 29A 4.6.3, Thornwood, NY, USA) [19,23,30,130]. Two examiners blinded to the *S. gordonii* infection and sham infection were measured the ABR.

### 4.5. Total RNA Isolation and Quality Assessment

Total RNA was extracted from left mandibles using the mirVana^TM^ miRNA Isolation Kit, following the manufacturer’s instructions, as described previously (Ambion, Austin, TX, USA) [19,23,30,130]. The final RNA yield and purity was assessed using spectrophotometric measurements. RNA samples exhibiting an OD 260/230 ratio of >2 and an OD 260/280 ratio of >2 were considered of sufficient quality for further NanoString (NanoString Technologies, Seattle, WA, USA) analysis.

### 4.6. NanoString nCounter miRNA Panel Analysis

For transcriptome analysis, total RNA extracted from the left mandibles was analyzed using NanoString nCounter^®^ Mouse miRNA Assay kit v1.5, a high-throughput nCounter^®^ miRNA Expression Panels to identify differential expression (DE) of miRNAs in 8- and 16-week *S. gordonii*-infected mice and sham-infected mice. Similar NanoString analysis was performed in polymicrobial [19], *P. gingivalis* [30], *T. denticola* [25], and *T. forsythia*-induced periodontal infections [23]. The NanoString nCounter^®^ panel is capable of simultaneously identifying and quantifying up to 577 miRNAs in a single RNA specimen by using molecular barcodes, called nCounter reporter probes. The NanoString capacity to detect miRNAs without requiring reverse transcription or amplification [19]. Briefly, nCounter^®^ miRNA assay panel kit has sample preparation steps of annealing, ligation, and purification. The Master mix preparation and thermal cycler conditions are explained more in our recent publications. Reported probe and miRNA capture probes was added after the purification steps. The strip tubes were incubated at 65 °C for 18 h in the thermal cycler, and the samples were immediately processed for post-hybridization with an nCounter analysis system in the Molecular Pathology Core at the University of Florida. A separate reporter code count (RCC) file for each sample containing the count for each probe was downloaded and used for miRNA data analysis.

### 4.7. NanoString Data Analysis

Data obtained from the NanoString platform were analyzed using the dedicated software nSolver™ 4.0 (NanoString Technologies, Seattle, WA, USA). The codeset incorporated 577 mouse miRNAs and included 6 ligation controls, and five miRNA reference controls (ACTB, B2M, GAPDH, RPL19, RPLP0). Data normalization was performed using nSolver Analysis Software v4.0 (NanoString Technologies). RCC files were downloaded and imported into nSolver and imaging quality control (QC), binding density QC, positive control linearity QC and positive control limit of detection QC were carried out as recommended in the system QC parameters. The lanes were flagged when the percent field of view (FOV) registration was <75% for imaging QC, binding density was outside the range from 0.1 to 2.25 for binding density QC, and the positive control R2 value was <0.95. All 40 samples passed the QC, and no flag lanes were observed. Raw data were created after passing the QC. To reduce the background signal/noise, the background threshold count value was set to 52 and analyzed by taking an average of eight negative control probe counts from all 40 samples. Codeset content normalization parameters were chosen, and the data normalization was performed based on the top 100 miR genes expressed. The normalized factor was calculated based on the geometric mean values of miR genes expressed in each specimen [19,23,25,30].

### 4.8. Bioinformatics Analysis

All the miRNA normalized data were analyzed further using ROSALIND^®^ (https://rosalind.bio/ (accessed on 21 September 2023), with a HyperScale architecture developed by ROSALIND^®^, Inc. (San Diego, CA, USA) [131]. Fold changes (FC) in the miRNA genes were calculated based on the ratio of the difference in the means of the log-transformed normalized data to the square root of the sum of the variances of the specimens in the four groups. The Limma R library [132] was used to calculate FCs. The clustering of miRNA for the final heatmap of DE miRNA was performed using the Partitioning Around Medoids method with the fpc R library. MiRTarBase was used to investigate validated miRNA-target gene interactions (miRTarbase update 2022: an informative resource for experimentally validated miRNA-target interactions) [133]. Venn diagram for higher expression and dysregulated miRNAs in the 8-week and 16-week infection groups was drawn using Venny 2.1 [19,23,25,30].

#### 4.8.1. Kyoto Encyclopedia of Genes and Genomes (KEGG)

KEGG is an integrated database from the National Center for Biotechnology Information RefSeq and GenBank and KEGG orthology [134]. KEGG pathways were plotted using DIANA-miRPath v.3.0 database [135] using the MIMAT accession number with the threshold values of *p* < 0.05 and false discovery rate (FDR) correction applied to obtain unbiased empirical distribution using the Benjamin and Hochberg method [19,23,25,30]. KEGG interprets the data from high-throughput experimental technologies. KEGG explores the possible key regulatory pathways for the enrichment of DE genes between the sham infection and *S. gordonii* mice infection.

#### 4.8.2. Multiple Machine Learning (ML) Analysis

To further investigate the impact of *S. gordonii* on PD, we used ML algorithms to analyze the NanoString data from the infected mice at the 8- and 16-week time points as well as the dataset formed by combining the 8- and 16-week datasets. For our analysis, we used XGBoost version 1.7.6 and the Scikit-learn version 1.3.0 implementations of LR, SVC, MLP, and RFC. The ML models were created and executed using Python version 3.11.4. LR [136], SVC [137] and RFC [138] were chosen because, historically, they have been extensively studied and the underlying mathematics is well understood. MLP was chosen because recent advances in computing power and ML libraries (e.g., Scikit-learn) now permit deep neural networks to be more easily set up and executed [139]. Moreover, the application of the MLP model to novel domains is an active area of research. Finally, XGB was chosen because of its widespread use in biomedical research [140]. For instance, on February 14, 2024, a search on PubMed for ‘XGBoost’ found 3995 articles.

LR uses the natural log of the odds ratio to predict the outcome of a binary variable [136]. Typically, a threshold of 0.5 is to determine whether the outcome is positive (e.g., a value of 1) or negative (e.g., a value of 0). SVC classifies data by calculating a hyperplane that maximally separates the target classes in a high-dimensional space. MLP is a type of neural network that uses multiple interconnected nodes and layers to learn the relations between linear and non-linear data and predict outcomes using the learned relations. RFC is an ensemble ML method that builds multiple decision trees (i.e., a “forest of trees”) and predicts an outcome by polling the decisions of the constructed trees. XBG, like RFC, constructs multiple decision trees, but XBG enhances the RFC approach by incorporating gradient boosting.

To address implicit biases in the data, we randomly shuffled the order of the rows and columns before analysis. We used scikit-learn’s RandomizedGridSearchCV method to determine the hyperparameters for each ML model. The RandomizedGridSearchCV method was executed using between 2 and 11 cross-validation splits of the data, and the performance of each cross-validation split was evaluated using the LeaveOneOut method. A full list of the hyperparameters used for each ML model is available in Appendix A. The LeaveOneOut method was chosen because of the small number of samples available in the data. SHAP version 0.42.1 was used to obtain feature importance results [23,140,141]. The code for executing the ML models on the NanoString copy data is available on GitHub at (https://github.com/uflcod/miRNA-periodontal-disease, accessed on 15 December 2023) in the ‘notebooks’ directory. The notebooks for analyzing the *S. gordonii* NanoString data begin with the prefix ‘sg_’ (e.g., sg_randomforest_miRNA.ipynb, sg_xgboost_miRNA.ipynb).

### 4.9. Statistical Analysis

All the data in the graphs were presented as mean ± SEM. Ordinary two-way ANOVA with Tukey’s multiple comparison test with a single pooled variance was performed for ABR measurements to identify the statistical significance using Prism 9.4.1 (GraphPad Software, San Diego, CA, USA) [19,23,25,30,142]. A *p*-value of <0.05 was considered statistically significant. NanoString data were used to obtain the miRNA expression in each specimen as an FC value, and an FC of ±1 was considered significant. Identification of *S. gordonii*-induced significant differential gene expression was done based on two-tailed t-tests on the log-transformed normalized data that assumed unequal variance. The distribution of the t-statistics was calculated using the Welch–Satterthwaite equation for the degrees of freedom to estimate the 95% confidence intervals for the identified DE of miRNA between infection and sham-infection mice. The volcano plot was drawn using GraphPad Software [19,23,25,30].

## 5. Conclusions

Over the past few years, critical knowledge of oral microbe-induced miRNAs is developing rapidly during PD pathogenesis as well as from human PD with and without diabetes and obesity. This study’s findings (attachment/colonization, ABR) demonstrates that *S. gordonii* a commensal oral bacterium can cause full development of PD. Out of several miRs’ anomalous expression (up- and downregulation) during *S. gordonii* bacterium monoinfection, a few miRNAs such as miR-375, miR-1902, miR-34b-5p are critical novel biomarker miRNAs during PD pathology. Specifically, multi-functional regulator miR-375 altered expression during monoinfection with *T. denticola, T. forsythia, S. gordonii* and several other diseases and diverse cancers indicating its use in diagnosis and clinical application.

## Figures and Tables

**Figure 1 ijms-25-06217-f001:**
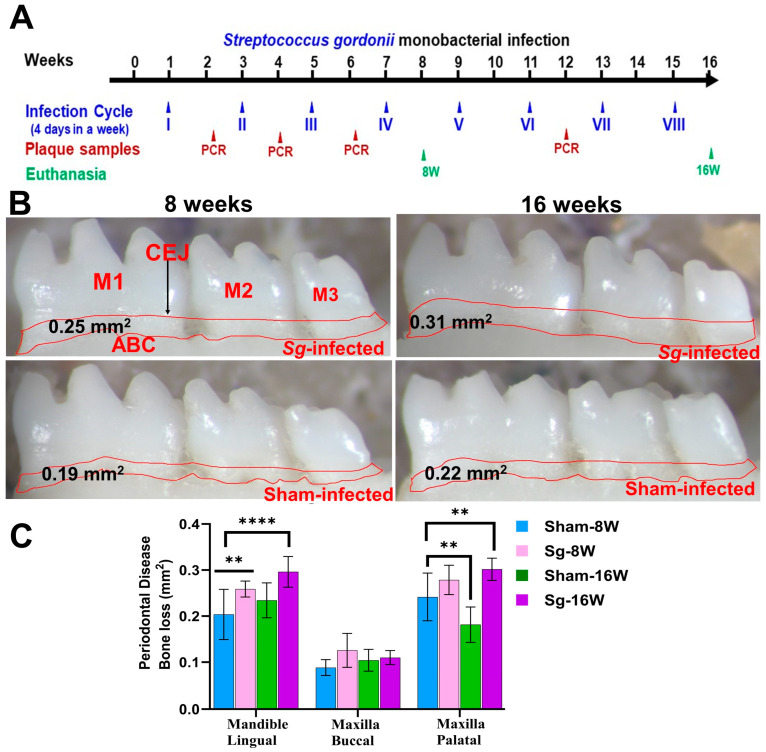
Intraoral infection of *S. gordonii* significantly induced ABR. (**A**) Schematic diagram of the experimental design depicting the monobacterial *S. gordonii* infection (4 days per week on every alternate week), plaque sampling for PCR, and euthanasia. (**B**) Representative images showing horizontal ABR (mandible lingual view) of *S. gordonii*-infected and sham-infected mice with the area of ABR outlined from the alveolar bone crest (ABC) to the cementoenamel junction (CEJ). (**C**) Morphometric analysis of the mandible and maxillary ABR in mice. A significant increase in ABR was observed in *S. gordonii*-infected mice compared to sham-infected mice at both 8-week and 16-week infected mice (** *p* < 0.05; ****, *p* < 0.0001; adjusted *p*-value = 0.0004; ordinary two-way ANOVA). Data points and error bars are mean ± SEM (*n* = 10).

**Figure 2 ijms-25-06217-f002:**
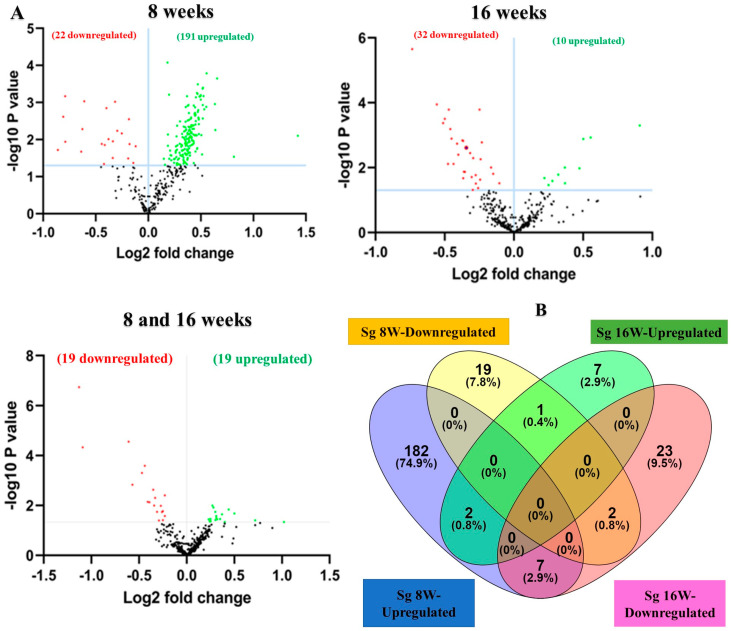
DE miRNAs in *S. gordonii*-infected mandibles (8- and 16-weeks). (**A**) The volcano plot depicts the upregulated (green) and downregulated (red) miRNAs that showed a fold difference of ±1.1 with a *p*-value of <0.05. The log2 fold change is on the *x*-axis, and the negative log of the *p*-value is on the *y*-axis. The black dots represent the miRNAs that do not pass the filter parameters. A total of 191 significant upregulated miRNAs and 22 downregulated miRNAs were identified in 8 weeks of *S. gordonii*-infected mice compared to 8 weeks of sham-infected mice (*n* = 10). Ten significant upregulated miRNAs and 32 downregulated miRs were identified in 16 weeks of *S. gordonii*-infected mice compared to 16-week sham-infected mice (*n* = 10). (**B**) Venn diagram analysis illustrates the distribution of DE miRs in 8-week and 16-week infections with *S. gordonii*. (**C**,**D**) Predicted functional pathway analysis of DE miRNAs from *S. gordonii*-infected mandibles. Bubble Plot of KEGG analysis on predicted target genes of DE miRNAs in *S. gordonii*-infected mice at 8- and 16 weeks of infection compared to sham-infected mice. The KEGG pathways are displayed on the *y*-axis, and the *x*-axis represents the false discovery rate (FDR), which means the probability of false positives in all tests. The size and color of the dots represent the number of predicted genes and corresponding *p*-value, respectively. Eleven DE miRNAs were shown to be involved in bacterial invasion of epithelial cells during 8 weeks of infection. Red color highlighted pathways explained in detailed in this manuscript.

**Figure 3 ijms-25-06217-f003:**
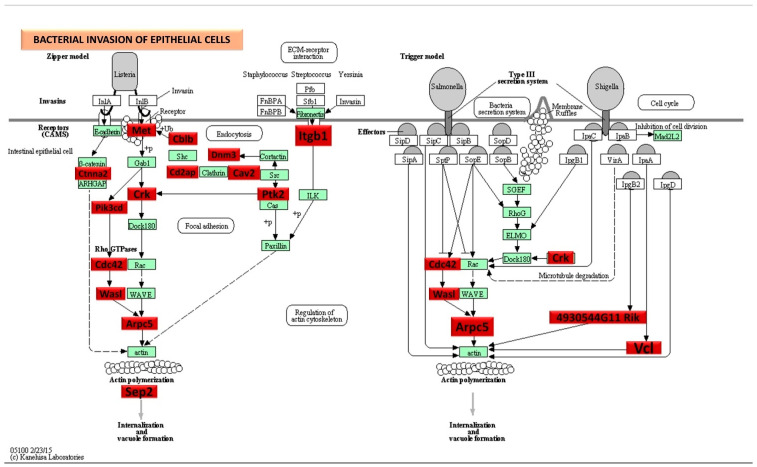
Significantly, DE genes (identified by KEGG) were involved in the bacterial invasion of epithelial cells signaling pathway during 8-week *S. gordonii* infection. Red boxes indicate 25 significantly increased expression based on the miRNA profiles from NanoString analysis. Green boxes indicate no change in gene expression. Many pathogenic bacteria can invade phagocytic and non-phagocytic cells and colonize them intracellularly, then become disseminated to other cells. Invasive bacteria induce their uptake by non-phagocytic host cells (e.g., epithelial cells) using two mechanisms referred to as the zipper model and trigger model. Listeria, Staphylococcus, Streptococcus, and Yersinia are examples of bacteria that can enter using the zipper model. These bacteria express proteins on their surfaces that interact with cellular receptors, initiating signaling cascades that result in close apposition of the cellular membrane around the entering bacteria. Shigella and Salmonella are examples of bacteria entering cells using the trigger model. An arrow indicates a molecular interaction and a line without an arrowhead indicates a molecular interaction resulting in inhibition.

**Figure 4 ijms-25-06217-f004:**
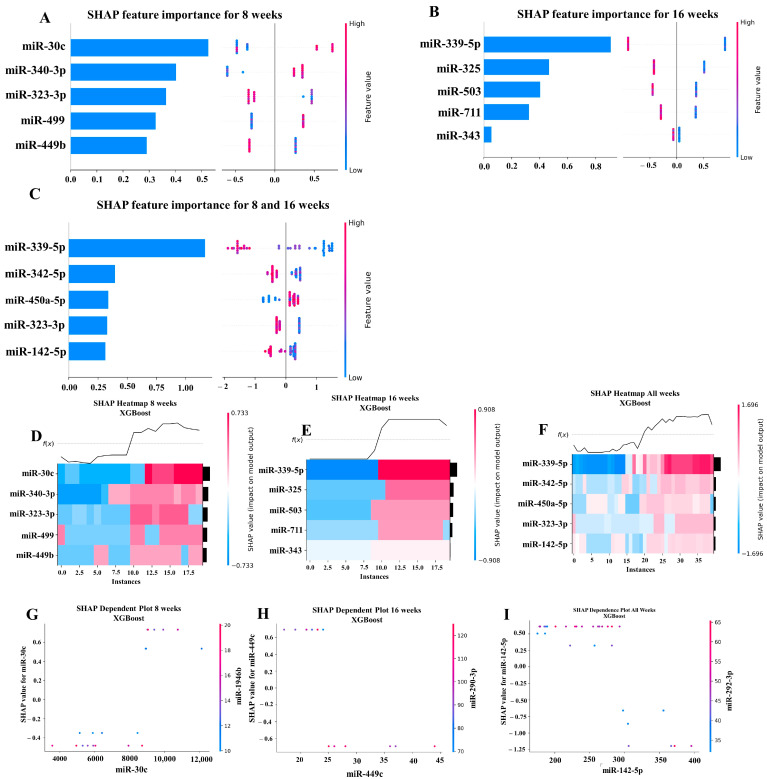
A summary of the most important features in the XGBoost model using SHAP values. In figures (**A**–**F**), feature importance is ranked from the top (the most important) to the bottom (the least important). The *x*-axis in (**A**–**C**) shows the impact that a feature has on the model. The bar charts show the overall impact of a feature whereas the swarm plot shows both the positive and negative impacts. In the swarm plots, each dot represents an instance of a miRNA variable and the color bar shows the value (high to low) of the variable. (**A**) Mice tested at 8 weeks. (**B**) Mice tested at 16 weeks are shown, and (**C**) 8- and 16-week cohorts together. (**D**–**F**) The *x*-axis represents each mouse (instance) in the cohort, and the *y*-axis is the feature ranking. The color of the cell shows the amount of impact (i.e., the SHAP value) that particular feature inflicted on that feature. The topmost section (i.e., f(x)) of the heatmap shows the predicted infection status for each instance. In all three cohorts, there is a strong correlation between high values for the topmost feature and the model predicting that the mouse was infected. In (**G**–**I**), the relation between the topmost feature and the feature it most depends on is shown. Each dot represents a mouse and the *x*-axis shows the value of the miRNA variable. The left *y*-axis shows the impact (i.e., the SHAP value) the *x*-axis variable has on the mode. The right *y*-axis shows the value of the variable that the *x*-axis variable interacts with.

**Figure 5 ijms-25-06217-f005:**
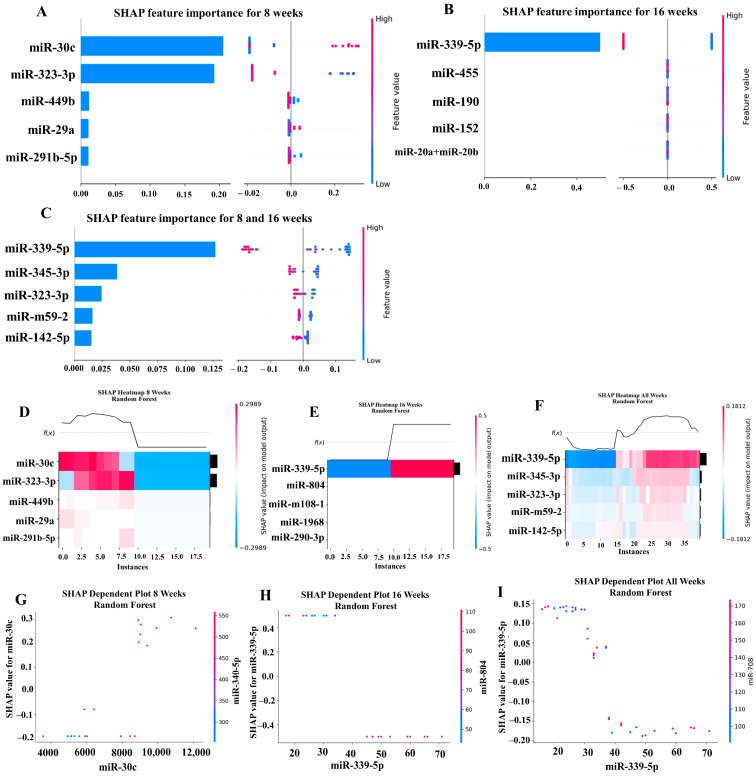
A summary of the most important features in the random forest classifier model using SHAP values.

**Figure 6 ijms-25-06217-f006:**
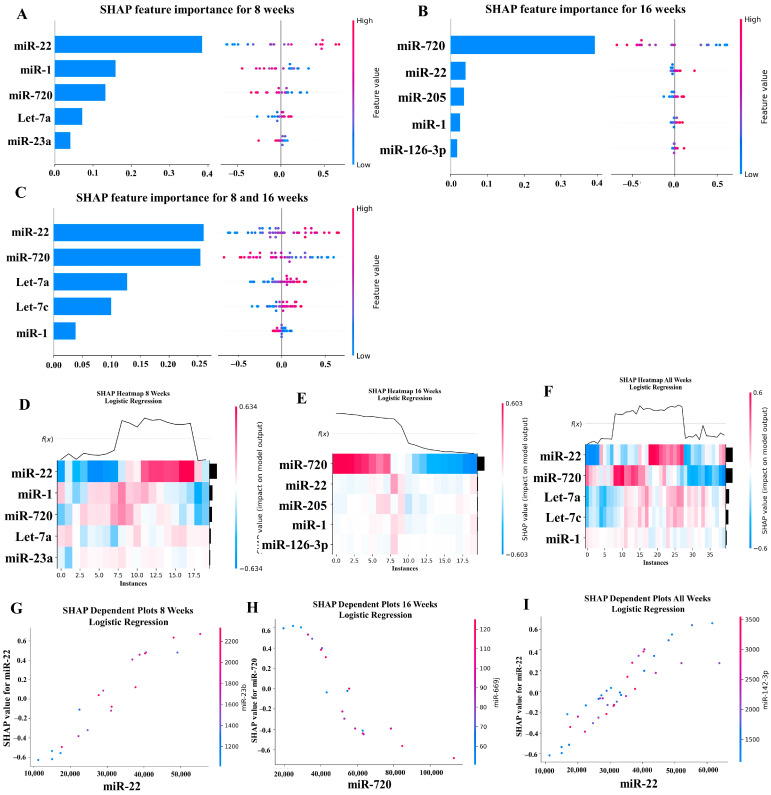
A summary of the most important features in the logistic regression model using SHAP values.

**Figure 7 ijms-25-06217-f007:**
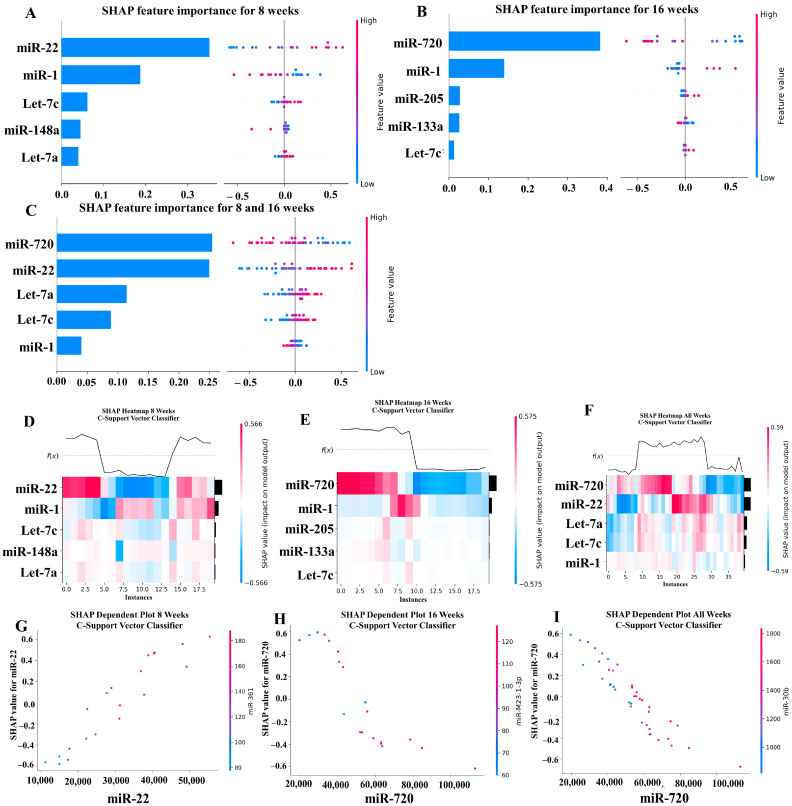
A summary of the most important features in the C-Support Vector Classifier model using SHAP values.

**Figure 8 ijms-25-06217-f008:**
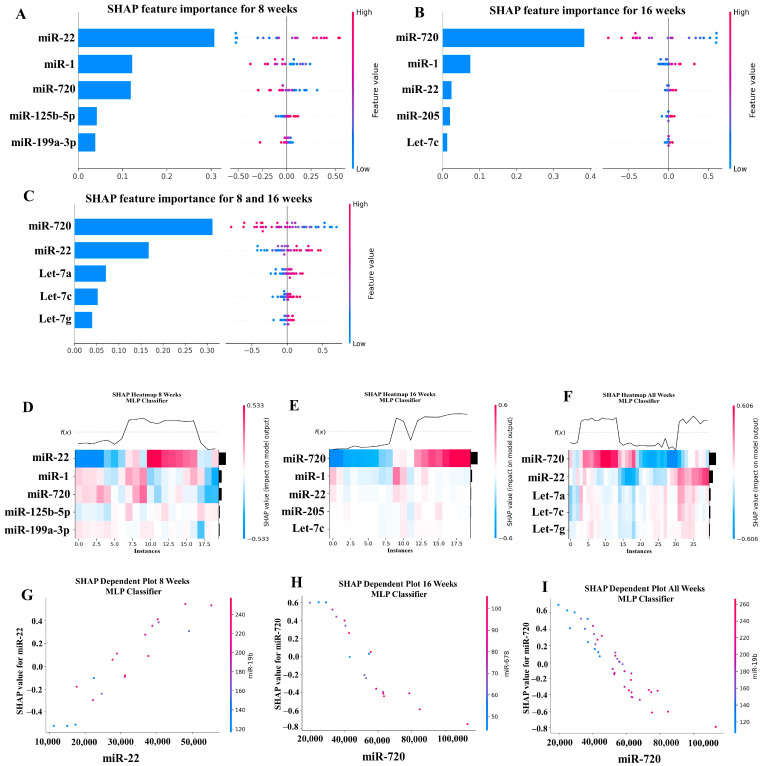
A summary of the most important features in the multilayer perceptron (Neural Network) model using SHAP values.

**Table 1 ijms-25-06217-t001:** Gingival plaque samples tested positive for *S. gordonii* gDNA using PCR.

Group/Bacteria/Weeks	Positive Gingival Plaque Samples (*n* = 10)			
	2 Weeks	4 Weeks	6 Weeks	12 Weeks
Group I/*S. gordonii* DL1 [8 weeks]	4/10a	9/10	NC	----
Group II/*S. gordonii* DL1 [16 weeks]	4/10	9/10	NC	10/10
Group III/Sham-infection [8 weeks]	0/10	NC	NC	----
Group IV/Sham-infection [16 weeks]	0/10	NC	NC	0/10

Total numbers of gingival plaque samples that were collected after *S. gordonii* infections (2, 4, 6, and 12 weeks) and were positive as determined by PCR analysis. NC—not collected to allow bacterial biofilm to adhere to the gingival surface, invade epithelial cells, and multiply. The first value corresponds to the number of mice that tested positive for *S. gordonii* bacterial genomic DNA, and the second value corresponds to the total number of mice in the group.

**Table 2 ijms-25-06217-t002:** Differentially expressed (DE) miRNAs during 8- and 16-weeks of *S. gordonii* infection.

Weeks/Infection/Sex	Upregulated miRNAs (*p* < 0.05)	Downregulated miRNAs (*p* < 0.05)
8 Weeks—*S. gordonii*-infected vs. 8 Weeks—Sham infection (*n* = 10)	191 (miR-375, miR-34b-5p, miR-142-5p, miR-135a).	22 (miR-133a, miR-1224, miR-2135, miR-499)
8 Weeks—*S. gordonii*-infected Female vs. Male (*n* = 5)	4	4
16 Weeks—*S. gordonii*-infected vs. 16 Weeks—Sham infection (*n* = 10)	10 (miR-1902, miR-203, miR-210, miR-876-3p)	32 (miR-720, miR-1937c, miR-2135, miR-326)
16 Weeks—*S. gordonii*-infected Female vs. Male (*n* = 5)	12	11
8 Weeks—*S. gordonii*-infected vs. 16 Weeks—*S. gordonii*-infected	19	19

The number of DE miRNAs was shown for *S. gordonii*-infected mice after 8- and 16-week infections. The commonly expressed miRNAs between 8- and 16-week bacterial infected groups are shown in brackets. Most of the miRNAs expressed in the bacterial-infected group were unique and specific to the 8- and 16-week infections.

## Data Availability

The data that support the findings of this study are openly available in NCBI https://www.ncbi.nlm.nih.gov/geo/query/acc.cgi?acc=GSE240502 (accessed on 15 August 2023).

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
