# Peer review of "Streptococcus gordonii Supragingival Bacterium Oral Infection-Induced Periodontitis and Robust miRNA Expression Kinetics"

_ijms, 2024, doi:10.3390/ijms25116217_

Round 1

Reviewer 1 Report

Comments and Suggestions for Authors

Few Methodological Biases exist

Many References are missing

(The Authors must see my remarks)

Author Response

Reviewers Comments:

REVIEWER 1:

Open Review

Quality of English Language

(x) I am not qualified to assess the quality of English in this paper
( ) English very difficult to understand/incomprehensible
( ) Extensive editing of English language required
( ) Moderate editing of English language required
( ) Minor editing of English language required
( ) English language fine. No issues detected

Yes

Can be improved

Must be improved

Not applicable

Does the introduction provide sufficient background and include all relevant references?

( )

(x)

( )

( )

Are all the cited references relevant to the research?

(x)

( )

( )

( )

Is the research design appropriate?

( )

(x)

( )

( )

Are the methods adequately described?

(x)

( )

( )

( )

Are the results clearly presented?

(x)

( )

( )

( )

Are the conclusions supported by the results?

(x)

( )

( )

( )

Comments and Suggestions for Authors:

  1. Few Methodological Biases exist: RESPONSE: To best of our knowledge, we have not observed any methodological bias.
  2. Many References are missing: RESPONSE: We have added all the references requested/embedded in the pdf.

(The Authors must see my remarks): RESPONSE: We have noted the remarks and addressed accordingly.

Reviewer 2 Report

Comments and Suggestions for Authors

Dear authors,

Thank you for submitting your valuable work to the journal. The topic of your in vitro research is interesting and could bring considerable new insights into the pathogenesis of periodontal disease. As generally considered that periodontitis is triggered by late colonizers, your resutls highlight that early colonizers could also have a significant impact on periodontal health, thus emphasizing even more the need of good oral hygiene habits. 

The paper is generally well writtent and its scientific background is sound. However, there are some comments I would make in order to increase its accuracy and improve accessbility to interested readers:

- please add a null hypothesis to the objectives of your study

- the sampling of subgingival biofilm could have been more relevant to the topic, why was only gingival plaque sampled? please explain

- tables 3 and 4 may seem redundant for the purpose of the paper, they should be included as supplementary files

- subsection 2.5 (MLA) should be partly moved to the Materials and Methods section, and only the results kept in section 2

- the limitations of the study should be more clearly discussed

We look forward to receiving the revised version of your manuscript.

Kind regards. 

Comments on the Quality of English Language

Minor editing

Author Response

Reviewers Comments:

Reviewer 2:

Open Review

Quality of English Language

( ) I am not qualified to assess the quality of English in this paper
( ) English very difficult to understand/incomprehensible
( ) Extensive editing of English language required
( ) Moderate editing of English language required
(x) Minor editing of English language required
( ) English language fine. No issues detected

Yes

Can be improved

Must be improved

Not applicable

Does the introduction provide sufficient background and include all relevant references?

( )

(x)

( )

( )

Are all the cited references relevant to the research?

(x)

( )

( )

( )

Is the research design appropriate?

(x)

( )

( )

( )

Are the methods adequately described?

( )

(x)

( )

( )

Are the results clearly presented?

( )

(x)

( )

( )

Are the conclusions supported by the results?

(x)

( )

( )

( )

Comments and Suggestions for Authors

Dear authors,

Thank you for submitting your valuable work to the journal. The topic of your in vitro research is interesting and could bring considerable new insights into the pathogenesis of periodontal disease. As generally considered that periodontitis is triggered by late colonizers, your results highlight that early colonizers could also have a significant impact on periodontal health, thus emphasizing even more the need of good oral hygiene habits. 

The paper is generally well written and its scientific background is sound. However, there are some comments I would make in order to increase its accuracy and improve accessibility to interested readers:

- please add a null hypothesis to the objectives of your study: RESPONSE: We are not comfortable to include null hypothesis in the objectives of study. We have published 4 papers in IJMS and did not include null hypothesis.

- the sampling of subgingival biofilm could have been more relevant to the topic, why was only gingival plaque sampled? please explain: RESPONSE: To best of our knowledge, there is no free gingiva in mouse and the extremely tightly bound tissue of the gingivae would not expand to accommodate to insert periopaper or cotton swab to take subgingival samples nor to inject even few microliters. We have published more than 20 papers with gingival plaque sampling only.

- tables 3 and 4 may seem redundant for the purpose of the paper, they should be included as supplementary files: RESPONSE: The tables 3 and 4 are for upregulated miRNAs for 8 and 16 weeks and these tables showing the miRNAs differentially expressed in mice which are very essential to the manuscript.

- subsection 2.5 (MLA) should be partly moved to the Materials and Methods section, and only the results kept in section 2: RESPONSE: We have revised as suggested by the reviewer.

- the limitations of the study should be more clearly discussed: RESPONSE: We have added a limitation for this study.

We look forward to receiving the revised version of your manuscript.

Kind regards. 

Comments on the Quality of English Language

Minor editing: RESPONSE: Revised the manuscript.

Round 2

Reviewer 2 Report

Comments and Suggestions for Authors

Dear authors,

Thank you for submitting the revised version of your manuscript. I have understood the rationale behind you explanations and I agree with the applied changes. I have no further comments.

Kind regards.